# Efficient RL-based Cache Vulnerability Exploration by Penalizing Useless Agent Actions

Kanato Nakanishi*, Soramichi Akiyama*

*Ritsumeikan University

*s-akym@fc.ritsumei.ac.jp*

*Abstract*—Cache-timing attacks exploit microarchitectural characteristics to leak sensitive data, posing a severe threat to modern systems. Despite its severity, analyzing the vulnerability of a given cache structure against cache-timing attacks is challenging. To this end, a method based on Reinforcement Learning (RL) has been proposed to automatically explore vulnerabilities for a given cache structure. However, a naive RL-based approach suffers from inefficiencies due to the agent performing actions that do not contribute to the exploration. In this paper, we propose a method to identify these useless actions during training and penalize them so that the agent avoids them and the exploration efficiency is improved. Experiments on 17 cache structures show that our training mechanism reduces the number of useless actions by up to 43.08%. This resulted in the reduction of training time by 28% in the base case and 4.84% in the geomean compared to a naive RL-based approach.

*Index Terms*—Reinforcement Learning, Side-Channel Attack, Cache-Timing Attack

## I. INTRODUCTION

As computer systems process more sensitive data, security becomes a critical concern. Among many threats, cache-timing attacks pose a serious risk. They exploit small differences in cache access times to leak sensitive data and leave minimal traces, making them hard to detect.

Reinforcement Learning (RL) can discover attack sequences for cache-timing attacks without prior knowledge. Here, an attack sequence is a series of actions taken by an attacker to leak sensitive data. Actions include reading data from a memory address and measuring its latency, flushing a cache line, and allowing the victim program to execute. AutoCAT [12] is an RL-based framework that automatically discovers successful attack sequences for a given cache structure (e.g., the number of sets and ways).

Although AutoCAT demonstrates potential, its naive RL approach introduces inefficiencies. We identified that many exploratory actions taken by AutoCAT fail to generate any meaningful change in the environment. In the context of cache-timing attacks, actions not altering the cache states do not contribute to exploration, because unchanged cache states would result in the same observations. We refer to such actions as *useless* actions throughout this paper. Useless actions increase the training time without contributing to the learning process.

This paper proposes a method to identify useless actions and reduce them, thereby improving exploration efficiency. We show that up to 43.08% of actions in a naive RL-based approach are classified as useless. We then introduce a penalty

mechanism that guides the agent toward more informative actions. Experiments on 17 cache configurations demonstrate up to a 28% reduction in the training time.

## II. BACKGROUND

### A. Cache-Timing Attacks

The goal of a cache-timing attack is to infer memory addresses that the victim process accesses. In this attack, the attacker is assumed to have an ability to run an arbitrary program with the user-level privilege on a CPU core that shares cache memory with the one executing the victim process. As the easiest example, suppose the cache is set-associative and has 4 sets. If the attacker accesses data at address 0 at time $T$ and $T+t$ and the latter access missed the cache (which can be known by the access latency), it can be inferred that the victim has accessed a data at address $A$ where $A$ mod 4 is 0 during the time period $t$. More sophisticated examples can be found in existing work such as Flush+Reload [20], Prime+Probe [11], and Evict+Reload [13].

By inferring memory addresses that the victim process accesses, it is possible to steal higher-level information such as cryptographic keys [20]. This is because many cryptographic algorithms access memory locations that are dependent on the cryptographic key. For example, block ciphers perform S-box lookups at table indices determined by specific bits in the key. An S-box (substitution box) is a fixed lookup table commonly used in block ciphers to introduce non-linearity. It maps an $n$-bit input to an $m$-bit output, with the specific table contents defined by the cipher's specification. By repeating cache-timing attacks over all possible S-box entries, it is known that an attacker can reconstruct the whole encryption key.

### B. Non-ML Approaches and Limitations

Prior work have addressed side-channel attacks (a superset of cache-timing attacks) by leveraging known attack patterns or specific hardware characteristics. Deng et al. [15] introduced a benchmark suite focusing on 88 known cache-timing vulnerabilities. He and Lee [8] proposed a methodology for quantitatively evaluating the resilience of cache structures against various cache-based attacks. Xiao et al. [19] and Buiras et al. [3] developed verification frameworks each tailored to particular hardware behaviors or predefined threat models. CheckMate [16] detects hardware vulnerabilities by synthesizing attacking programs from predefined patterns.

Despite these efforts, conventional methods for analyzing cache-timing attacks face two major challenges. First, Modern processors employ complex, proprietary cache designs. Key features like replacement policies and prefetching algorithms are rarely documented [2], [4], [17], forcing researchers to reverse engineer such internals [18]. Second, the search space for potential attack sequences is vast, making manual or pattern-based analysis inefficient and incomplete.

### C. Applying RL for Cache-Timing Attack Exploration

RL offers a promising solution to the challenges described in Section II-B because it can explore diverse attack sequences automatically. As long as an interface for actions and observations is defined, RL can explore attack sequences regardless of the underlying cache structures.

AutoCAT [12] is the first framework to use RL for exploring cache-timing attacks. In AutoCAT, an attacker is modeled as an RL agent and the environment consists of the victim process and the cache system. It uses a deep RL approach where the action value function is represented by a neural network. The network receives a vector representation of the observation and past actions, and then outputs the expected total reward the agent will collect if it selects each action. The actions are reading a memory location and measuring its latency, flushing a specific cache line, guessing which address the victim accessed, and allowing the victim to execute. The environment then calculates the outcome of the chosen action and provides two types of feedback. First, it returns an observation, indicating whether the agent's access resulted in a cache hit or miss. Second, it provides a reward signal reflecting how successful the action was. In concrete, the agent receives a small negative reward after an action as a penalty for prolonging the attack sequence, and a large positive reward when it successfully guesses the secret address.

Training in AutoCAT is organized as a set of epochs, each of which is a set of episodes. An episode starts with a full reset of the environment (e.g., clearing the cache) and ends with a guess action by the agent. At the beginning of every episode, the environment randomly selects a secret address that the victim will access. An epoch ends after 3,000 actions, even if the current episode is still in progress. After each epoch, the *correct rate* is computed. The correct rate is the fraction of episodes during the epoch in which the agent has correctly guessed the secret address.

### D. Inefficiencies in Naive RL Approach

Although promising, AutoCAT's naive RL approach leads to significant inefficiencies. In our experiments, a single training run of AutoCAT [6] on a machine equipped with dual Intel Xeon Platinum 8380 CPUs (40 cores each), 512 GB of DDR4 3200 memory, and an NVIDIA RTX A4500 GPU takes over two hours. This is impractical for two reasons:

1) AutoCAT discovers only one attack sequence per training run by design. Therefore, multiple runs are required to uncover a comprehensive set of vulnerabilities. In fact,

several fundamentally different attack sequences were found for the same cache structure in our experiments.

2) Functional verification of IC and ASIC designs consumes the majority of development schedules. According to the 2020 Wilson Research Group study [7], an average of 56% of IC and ASIC project time was spent on functional verification. It also reports that 68% of the projects fall behind the original schedules. Therefore, little time remains for security validation such as side-channel vulnerability testing.

We identify useless agent actions as a major reason for this inefficiency. We define an action as *useless* when it does not contribute to further exploration of the RL agent. In our context, this corresponds to actions that do not alter the cache states after its execution. Below are typical cases.

1) **Accessing data that is already cached**: This does not contribute to further exploration because observations by any action following this action do not change.

2) **Flushing a cache line that is not currently cached**: For the same reason, this does not contribute to further exploration.

### III. RELATED WORK

MACTA [5] improves the stealthiness of attack sequences, meaning that it generates ones that are less likely to be detected. It achieves this by adopting a dual-policy framework, where the attacker and detector policies are alternately optimized in a fictitious-play loop. Through iterative updates of both policies, MACTA produces attack sequences that not only remain effective on real hardware but also evade the trained detector with over 99% success rate.

Although important itself, MACTA does not address the inefficiencies we identified. This is because MACTA's reward function emphasizes primarily attack success and detection evasion and only applies penalties based on sequence length. Therefore, it does not explicitly penalize actions that do not change the cache state and its actions could include useless ones. We focus explicitly on enhancing the efficiency of the RL exploration by penalizing actions that leave the cache state unchanged, rather than improving the stealthiness of attacks or co-training a detector.

As summarized in Section II-B, non-ML approaches rely on known attack patterns or specific hardware characteristics [2], [4], [16], [17], lacking RL's ability to explore automatically. As a result, they struggle to handle undocumented cache behaviors and to efficiently navigate the vast search space of potential attack sequences.

### IV. PROPOSED METHOD

#### A. Overview

In this work, we propose a new RL-based method for cache vulnerability exploration that is more efficient than the naive approach. We achieve this by detecting useless agent actions and training the agent to avoid useless actions in future exploration. Our approach comprises two key techniques:

1) Mechanism to identify useless actions
2) Training mechanism that drives the agent to avoid useless actions during exploration

We explain each of them in the following subsections.

### B. Identifying Useless Actions

To systematically identify useless actions, we compare the cache states before and after each action. This alleviates the need to extensively elaborate all the possible cases of useless actions (and possibly missing some corner cases).

Capturing cache states to compare them involves different methods depending on the environment. When the environment is a cache simulator, the cache states can be captured from the in-memory representation of the simulated cache. When the environment is real hardware, it is not as straightforward because probing the cache states by some measures (e.g., accessing some data) itself might change them. Establishing a plausible way for this is our future work.

### C. Training the Agent to Avoid Useless Actions

One possible solution of reducing the useless agent actions is to introduce an ad-hoc mechanism in which the agent records its past actions and ensures that the next action is not useless. For example, we can let the neural network output an action normally, and then avoid it (choose a different one) if it is useless. However, this method lacks generalizability and does not fully utilize the RL framework.

To overcome this limitation, we leverage the reward signals of the RL framework. Specifically, we assign an immediate negative reward whenever the agent executes a useless action. This way we can tell the agent to avoid useless actions without ad-hoc implementation.

In concrete, our training mechanism works as follows:

1) **Capture the current cache state**: Before executing an action, the environment captures the current cache states.
2) **Execute the action**: The agent performs the chosen action normally.
3) **Compare cache states**: After the action, the environment captures the new cache states and compares them to the previous states. If the cache states have not changed, the action is classified as useless.
4) **Assign a negative reward**: Actions identified as useless are assigned a small negative reward (e.g., $-0.01$).

### D. Implementation

We use the public AutoCAT implementation [6] as our baseline and extend its environment in two aspects. First, we add logic that determines whether an action has changed the cache states to identify useless actions. The cache states are captured by calculating the hash value of the in-memory object that represents the simulated cache. Therefore, our current prototype only works for simulated caches. Second, we modify the rewards so that the agent receives a small negative reward whenever it performs such a useless action. The actual value of the negative reward is written in a config file and can be easily adjusted. Actions that allow the victim to execute and

TABLE I
TESTED CACHE CONFIGURATIONS

| Config. | Type | Ways | Sets | Victim Addr | Attacker Addr | Flush Inst |
|---|---|---|---|---|---|---|
| No.1 | DM | 1 | 4 | 0~3 | 4~7 | no |
| No.2 | DM+PFnextline | 1 | 4 | 0~3 | 4~7 | no |
| No.3 | DM | 1 | 4 | 0~3 | 0~3 | yes |
| No.4 | DM | 1 | 4 | 0~3 | 0~7 | no |
| No.5 | FA | 4 | 1 | 0 | 4~7 | no |
| No.6 | FA | 4 | 1 | 0 | 0~3 | yes |
| No.7 | FA | 4 | 1 | 0 | 0~7 | no |
| No.8 | FA | 4 | 1 | 0~3 | 0~3 | yes |
| No.9 | FA | 4 | 1 | 0~3 | 0~7 | yes |
| No.10 | DM | 1 | 8 | 0~7 | 0~7 | yes |
| No.11 | FA | 8 | 1 | 0 | 0~7 | yes |
| No.12 | FA | 8 | 1 | 0 | 0~15 | no |
| No.13 | FA+PFnextline | 8 | 1 | 0 | 0~15 | no |
| No.14 | FA+PFstream | 8 | 1 | 0 | 0~15 | no |
| No.15 | SA | 2 | 4 | 0~3 | 4~11 | no |
| No.16 | 2-level SA | 2 | 4 | 0~3 | 4~11 | no |
| No.17 | 2-level SA | 2 | 8 | 0~7 | 8~23 | no |

guess the secret are exempt from the negative reward because they are not expected to change the cache states.

## V. EVALUATION

### A. Evaluation Setup

We evaluate our method by comparing it with a naive RL-based approach. The vanilla AutoCAT implementation [6] is used as the baseline for the comparison. Both our system and vanilla AutoCAT use an open-source cache simulator [1]. This simulator supports multiple replacement policies, including LRU, random [10], PLRU [14], and RRIP [9]. The simulated cache is physically indexed and tagged. Both the attacker and the victim access main memory with physical addresses.

The training hyperparameters we use are as follows. First training continues until either the agent discovers an attack sequence with 100% correct rate or 999 epochs are completed. If the agent fails to reach this accuracy within 999 epochs, we consider the training non-convergent. The other parameters are the same as the vanilla AutoCAT. The learning algorithm is PPO. The neural networks are updated every 2048 actions, split into minibatches of 64.

- Hidden layer: 256 units with ReLU activation function
- Optimizer: Adam
- Discount factor ($\gamma$): 0.99
- Learning rate: $3 \times 10^{-4}$
- PPO clip ratio: 0.2

We conduct our experiments for 17 cache structures shown in Table I. For each cache structure, experiments were conducted 10 times and the average result is presented. We vary the number of sets and ways, and toggle the availability of cache flush instructions. DM, FA, and SA represent direct-mapped, fully-associative, and set-associative caches, respectively. Some configurations include prefetchers, such as the nextline prefetcher or the stream prefetcher. Configurations labeled "2-level SA" model a two-core system with a two-level cache hierarchy. Each core has its own private direct-mapped L1 cache, and both cores share a single inclusive set-associative L2 cache.

## TABLE II
### REDUCTION OF USELESS ACTION RATIO

| Config. | Approach | Total Actions | Useless Action Ratio (%) | Delta (pts) |
|---------|----------|---------------|--------------------------|-------------|
| No.1 | Baseline | 1,278,786 | 32.79 | – |
|  | Proposal | 1,561,594 | 39.62 | +6.83 |
| No.2 | Baseline | 1,403,410 | 33.13 | – |
|  | Proposal | 1,393,471 | 26.32 | **-6.81** |
| No.3 | Baseline | 2,812,834 | 30.09 | – |
|  | Proposal | 3,160,743 | 27.82 | **-2.27** |
| No.4 | Baseline | 2,436,943 | 33.87 | – |
|  | Proposal | 2,410,048 | 31.27 | **-2.60** |
| No.5 | Baseline | 2,988,773 | 38.49 | – |
|  | Proposal | 2,081,238 | 46.70 | +8.21 |
| No.6 | Baseline | 1,325,707 | 34.43 | – |
|  | Proposal | 2,368,477 | 24.83 | **-9.60** |
| No.7 | Baseline | 3,041,778 | 29.12 | – |
|  | Proposal | 3,545,075 | 27.66 | **-1.46** |
| No.8 | Baseline | 3,145,930 | 34.59 | – |
|  | Proposal | 3,361,129 | 36.97 | +2.38 |
| No.9 | Baseline | 2,530,719 | 35.61 | – |
|  | Proposal | 5,031,057 | 32.49 | **-3.12** |
| No.10 | Baseline | 44,111,128 | 16.32 | – |
|  | Proposal | Non-convergent | – | – |
| No.11 | Baseline | 2,160,143 | 37.66 | – |
|  | Proposal | 2,006,860 | 36.81 | **-0.85** |
| No.12 | Baseline | 11,392,617 | 30.71 | – |
|  | Proposal | 4,259,088 | 30.94 | +0.23 |
| No.13 | Baseline | 12,687,713 | 37.22 | – |
|  | Proposal | 20,920,890 | 28.20 | **-9.02** |
| No.14 | Baseline | 4,406,989 | 36.75 | – |
|  | Proposal | 4,916,277 | 32.97 | **-3.78** |
| No.15 | Baseline | 6,117,652 | 43.08 | – |
|  | Proposal | 10,915,735 | 32.69 | **-10.39** |
| No.16 | Baseline | 8,221,525 | 22.10 | – |
|  | Proposal | 5,379,145 | 24.28 | +2.18 |

We evaluate our method with the following two metrics:

1) **Reduction of Useless Action Ratio:** We compare the proposed method with the baseline to assess the difference in useless action rates.
2) **Reduction of Total Training Time:** We compare the proposed method with the baseline to assess the difference in total training time.

### B. Reduction Rate of Useless Agent Actions

Table II shows the ratio of useless actions for the baseline and for our proposed method. The result for configuration No. 17 is not shown because the training was non-convergent. Overall, the results suggest that a considerable number of the agent's actions are useless in the baseline. Furthermore, the proportion of these useless actions differs across various cache configurations. In particular, configuration No. 15 experienced the highest ratio of 43.08% while configuration No. 10 showed the lowest ratio of 16.32%.

Applying our proposal reduces the ratio in 9 out of 16 convergent setups. For instance, the ratio decreases from 33.17% to 26.35% in configuration No. 2 and from 34.49% to 24.85% in configuration No. 6. In contrast, some configurations exhibit an increase in the useless action ratio. For example, it increases by 6.85 points in configuration No. 1 and by 8.21 points in configuration No. 5.

Overall, the results indicate that our proposal effectively reduces useless actions in many scenarios. However, its impact depends on the cache configuration. The results suggest that our mechanism may unintentionally penalize useful actions in certain cache configurations.

### C. Reduction of Total Training Time

Fig. 1 presents the training time results for the 17 cache configurations. To improve readability, the figure is split into two separate bar charts. The upper chart shows the results for configurations No. 1 through No. 9 and No. 11 through No. 16. Configurations No. 10 and No. 17 are shown in a separate lower chart with a different vertical axis scale because their training times are much longer than the others.

Our proposal reduced the total training time in 11 cache configurations out of 17. The geometric mean of the training time ratio against the baseline across the 16 convergent configurations (No. 1 to No. 16) was 0.9516 and corresponds to an average reduction of 4.84%. In the best cases, we observed a 28% speedup compared to the baseline in configuration No. 11 and 27% speedup in configuration No. 14. However, in the worst case, we observed a 57% slowdown compared to the baseline in configuration No. 10.

Our hypothesis for configuration No. 11 being reduced the total training time the most is threefold.

1) The agent sometimes issues a series of flush actions at the start of each episode to ensure that the cache is empty. For example, the agent may generate the following actions: flush 0 $\rightarrow$ flush 1 $\rightarrow$ flush 2 $\rightarrow$ ... Here, an action "flush N" invalidates the cache line that contains data at address N.
2) However, many of these flushes do not contribute to exploration because the victim always accesses a single address. Therefore, if the secret address is 0, the initial flushes besides flush 0 do not contribute to the attack.
3) Our proposal penalized these flushes because they are useless in our definition.

In contrast, the total training time was increased the most in configuration No. 10. Our hypothesis for the reason is that actions contributing the attack are classified as useless due to our definition. Because the secret address spans from 0 to 7 in this configuration, the attacker must ensure that all the cache lines are empty before allowing the victim to execute. However, this confirming actions are classified as useless in our definition because flushing a cache line corresponding to a non-cached address does not change the cache states.

### D. Impact of Useless Action Reduction on Training Time

We analyze the relationship between the reduction in the useless action ratio reported in Table II and the total training time shown in Figure 1. Among the ten configurations where our proposal reduced the useless action ratio (No. 2, 3, 4, 6, 7, 9, 11, 13, 14, and 15), the total training time also decreased in nine of them (all except No. 15). However, there is no strong correlation between the reduction rate of the useless agent actions and the speedup of the training time. One of the

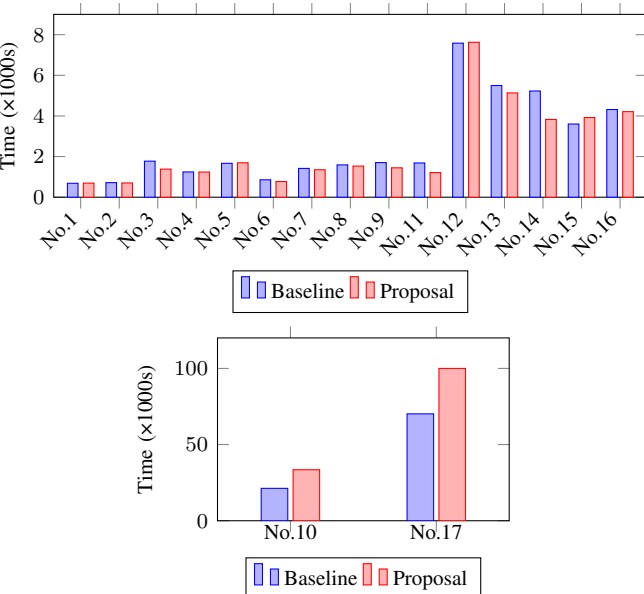

Fig. 1. Comparison of Total Training Time (Time in 1000s)

reasons behind this is that the total number of actions also changed under our proposal.

In configuration No. 15, the useless action ratio decreased by 10.39 percentage points but the training time increased by approximately 9%. This result can be attributed to the fact that the total number of actions increased from approximately 6.1 million in the baseline to 10.9 million under our proposal. This led to the longer training time in our proposal even though it achieved a lower useless action ratio.

## VI. CONCLUSION

Evaluating the vulnerabilities of a given cache structure to cache-timing attacks is an important and time-consuming task. To this end, we identified a major reason of a naive RL-based approach for cache vulnerability exploration is useless agent actions. Based on this observation, we proposed a novel method to train an RL agent to avoid useless actions to improve the training efficiency. Our evaluation revealed that that up to 43.08% of agent actions were useless on 17 configurations, showed that our proposal reduced them by up to 10.39 points and reducing the training time by 28% in the best case and by 4.84% in the geomean.

### ACKNOWLEDGEMENTS

This work was supported by JST, PRESTO Grant Number JPMJPR22P1, Japan. We thank the anonymous reviewers for their valuable feedback to improve this paper.

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
