# OpenReview forum: "Efficient RL-based Cache Vulnerability Exploration by Penalizing Useless Agent Actions"
_iscaconf.org/ISCA/2025/Workshop/MLArchSys — MLArchSys 2025 Oral_

### Official Review · Reviewer_m2ks · 2025-05-18
**Leveraging improved RL-based tools to penalize "useless" actions.**

**Confidence:** 3
**Rating:** 6

**Detailed Feedback And Questions For Authors:**

Overall, I thought this was an interesting paper. At a high level, it is doing pruning of the search space (or perhaps, domain search exploration) in the RL space to avoid going down unnecessary paths by the RL system. The application to cache time channels is also a practical use case. However, as with any ML based technique, it would be good to highlight drawbacks to strengthen the paper. As a reviewer, the concept of false negatives (avoiding paths that could potentially hold useful information) is important, and I did not fully capture that for the authors in this version.

I will note that one sentence in the paper really caught my attention: "When the environment is real hardware, it is not as straightforward because probing the cache states by some measures (e.g., accessing some data) itself might change them. Establishing a plausible way for this is our future work."
To me, this is a very very interesting future direction, and highly encourage exploring it further!

From an approach perspective: "The results suggest that our mechanism may unintentionally penalize useful actions in certain cache configuration" this seems like it could potentially be addressed by some sort of separation of concerns. I don't have a solid solution, but believe there could be a way from the ML literature to avoid hurting the reward function to a good degree.

**Top Reasons To Accept The Paper:**

This is a nice idea, focused on improving the efficiency of an RL-based mechanism for the application of cache timing attacks. The idea is well implemented and studied. Overall, a good research direction, although both the methodology and evaluation can be improved.

**Top Reasons To Reject The Paper:**

I'd be curious to understand the false negatives of the approach: is it possible that something gets classified as "useful", but it is in fact "useful"? How often does this happen? It was not so clear to me.

Also, if there were no resource constraints (e.g., plentiful A100s), how important would this problem be? Could the extra compute be helpful to mitigate false negatives, and quickly churn through various options?

Neither of these are true showstoppers - but considering them would strengthen the paper.

---

### Official Review · Reviewer_tJMs · 2025-05-19
**"Efficient RL-based Cache Vulnerability Exploration ..." : This work aims to improve efficiency of an RL based cache vulnerability framework by identifying/penalizing useless actions in the RL method.**

**Confidence:** 4
**Rating:** 6

**Detailed Feedback And Questions For Authors:**

I enjoyed reading this paper. I found that the topic and the presented technique are interesting and fit well with the workshop, although the key technical ideas and the performance gain looked a bit marginal.

Below are some other comments regarding the paper.
- On p1. S-box: the authors can add the definition for completeness.
- Table 2: the authors can add the definition of "Delta (pts)".
- Figure 1: the different scales of the Y-axis (0-10 and 10-100) might be confusing and/or not easy to compare some small numbers. I was wondering if normalizing/relative or log-scale numbers might be helpful.

**Top Reasons To Accept The Paper:**

- This work focuses on an interesting research problem on cache architecture vulnerability exploration. The presented work has the potential to reduce the security validation time in hardware design.

- The topic of this work fits very well with the scope of the workshop.

- The experiments are reasonably designed and evaluated on various cache architectures.

**Top Reasons To Reject The Paper:**

- The technical contributions are marginal. The main technical idea is to capture the changes of the cache and apply a simple generalized penalization method for useless actions to the baseline RL method. I wondered if there would be more structural attempts to help improve efficiency.

- From the experimental results, the improvements of the presented technique do not seem consistent across various cache architectures, with some quite longer training time in some complex cache settings.

---

### Official Review · Reviewer_WY68 · 2025-05-19
**This paper proposes new RL-based method to efficiently train and use an RL agent for analyzing cache vulnerability**

**Confidence:** 4
**Rating:** 5

**Detailed Feedback And Questions For Authors:**

Thank you for submitting your work to MLArchSys'25. I think the paper focuses on a real problem, motivates the need for ML (RL), and proposes a methodology to improve the state of the art.

My first major comment is about the evaluation: while the paper describes what is observed in the evaluation, I was hoping to understand the results more intuitively. For example, the paper describes that the observed proportion of the useless actions differs across various cache configurations. I was wondering if there were any insights to explain these differences, and what cache configuration is likely to benefit more versus not using the proposed methodology. Consider explaining these.

Secondly, I didn't understand the connection between the results in Table 1, showing a reduction in useless actions, and Figure 1, showing a comparison of training times. I expected that the reduction in useless actions would result in a reduction in training time, too. But, in some cases, this isn't true. For instance, in configuration No.1, while the proposed methodology ends up increasing the useless actions, the training time isn't affected. Consider explaining such cases.

**Top Reasons To Accept The Paper:**

- The paper is well-written and easy to follow
- The proposed methodology introduces a new way to efficiently explore the action space using an RL agent

**Top Reasons To Reject The Paper:**

- The insights behind the observed empirical results are missing.

---

### Official Review · Reviewer_q5YU · 2025-05-21
**RL for discovering cache-timing attacks with negative reward assignment for useless actions**

**Confidence:** 2
**Rating:** 6

**Detailed Feedback And Questions For Authors:**

This paper applies reinforcement learning to cache-timing attacks and introduces an automatic reward-shaping mechanism that penalizes unproductive actions with negative rewards.

For evaluation, the paper would be strengthened by reporting the geometric mean improvement in training time and by comparing against a baseline where unproductive actions are explicitly masked rather than penalized.

**Top Reasons To Accept The Paper:**

+ Interesting ML application to target
+ Reasonable reward assignment

**Top Reasons To Reject The Paper:**

- The current prototype only works for simulated caches
- Limited training time improvement
- Lack of comparison to masking useless actions